# Analysing Complex Oral Protein Samples: Complete Workflow and Case Analysis of Salivary Pellicles

**DOI:** 10.3390/jcm10132801

**Published:** 2021-06-25

**Authors:** Chen-Xuan Wei, Michael Francis Burrow, Michael George Botelho, W. Keung Leung

**Affiliations:** 1Faculty of Dentistry, The University of Hong Kong, Hong Kong, China; ivycxwei@umich.edu (C.-X.W.); mfburr58@hku.hk (M.F.B.); botelho@hku.hk (M.G.B.); 2School of Dentistry, University of Michigan, Ann Arbor, MI 48104, USA

**Keywords:** dental pellicle, mass spectrometry, protein, proteogenomic, saliva

## Abstract

Studies on small quantity, highly complex protein samples, such as salivary pellicle, have been enabled by recent major technological and analytical breakthroughs. Advances in mass spectrometry-based computational proteomics such as Multidimensional Protein Identification Technology have allowed precise identification and quantification of complex protein samples on a proteome-wide scale, which has enabled the determination of corresponding genes and cellular functions at the protein level. The latter was achieved via protein-protein interaction mapping with Gene Ontology annotation. In recent years, the application of these technologies has broken various barriers in small-quantity-complex-protein research such as salivary pellicle. This review provides a concise summary of contemporary proteomic techniques contributing to (1) increased complex protein (up to hundreds) identification using minute sample sizes (µg level), (2) precise protein quantification by advanced stable isotope labelling or label-free approaches and (3) the emerging concepts and techniques regarding computational integration, such as the Gene Ontology Consortium and protein-protein interaction mapping. The latter integrates the structural, genomic, and biological context of proteins and genes to predict protein interactions and functional connections in a given biological context. The same technological breakthroughs and computational integration concepts can also be applied to other low-volume oral protein complexes such as gingival crevicular or peri-implant sulcular fluids.

## 1. Introduction

The oral cavity serves as a window to the human body where observations made upon analysis of oral fluids could potentially reflect the health, disease or stressful psychological status of the person being investigated. For instance, saliva carries precise developmental and biological information that defines the function of multiple organs including oral and periodontal tissues [1]. Salivary pellicle, also known as acquired enamel pellicle, is an important interface structure/proteinaceous material located between dental hard tissue surfaces and the oral environment which modulates the biological and microbiological activities on tooth surfaces [2]. However, often, due to a lack of suitable experimental protocols, and small quantities of highly complex protein samples acquired from the oral cavity, such as salivary pellicle, limits the information that can be extracted from such samples. Advances in mass spectrometry-based computational proteomics since the past decade or so have allowed precise identification and quantification of complex protein samples in a relatively small sample volume and enabled the determination of detailed biological information such as corresponding genes and cellular functions at the protein level [3]. This review adopted salivary pellicle as an example to demonstrate a step by step workflow of protein sample collection, processing and analysis. The corresponding workflow could also be applied to other low-volume highly-complex oral protein sample analysis such as gingival crevicular or peri-implant sulcular fluids.

The salivary pellicle is a proteinaceous film, that forms within seconds after tooth eruption or prophylaxis [4], and is claimed to be highly selective and dynamic, and greatly affected by the local microenvironment and physiochemical properties of the substratum [4]. After initial attachment, the later arriving proteins with higher affinity could competitively replace earlier adsorbed proteins; this is known as the Vroman effect [5]. Lee et al. [6] illustrated the dynamic protein adsorption and exchange phenomenon on tooth surfaces. They detected 89, 92, 107 and 101 proteins from 5-min, 10-min, 1-hour and 2-hours in vivo enamel salivary pellicles, with only 50 proteins common at all four-time points. Despite compositional changes, different enamel salivary pellicle protein compositions were also observed at different locations of the dentition [7]. As the oral cavity is a unique portal where mineralized tissue is directly exposed to heavy colonization of oral microbiota, it is plausible that the acquired salivary pellicle interfacing between tooth surfaces, the oral environment, and intimate contacts with oral microbes could hold essential biological functions, such as protection from demineralization, buffering bacterial acids or other toxic by-products that may lead to dental caries initiation [8]. Therefore, the scientific approaches in understanding salivary pellicle proteins on tooth surfaces are of great importance in comprehending their roles and relationships with oral microbes and dental disease pathogenesis.

## 2. Salivary Pellicle Proteomics

The term “proteomics” refers to a global technique to study proteins. This review will summarize how state-of-the-art proteomic approaches can be employed to study low abundance protein complexes, such as salivary pellicle. Topics include sample preparation methods, protein separation, identification and quantification, and functional annotation. To complete the review, a brief overview of factors crucial to salivary pellicle sampling are also included.

### 2.1. Salivary Pellicle Protein Sample Preparation

Salivary pellicle proteins derived from saliva, are highly dynamic and variable due to the physiology and bioactivity of host salivary protein biosynthesis and post-secretion modification [9]. Hence, appropriate controls, inclusion and exclusion criteria, such as gender, age, circadian rhythm, medical history and other individual variables should be taken into consideration to minimize the heterogeneity among samples harvested from different individuals [10] when analyses are performed. When comparing between individuals, due to high intraindividual and interindividual variability of initial bioadhesion, it is also important to highlight the overlap seen in proteins that are involved in different pathways/biological processes, which may indicate a higher affinity, and potential biomarkers [11]. For instance, Winter et al. [12] employed state-of-the-art MS-based proteomics to obtain proteome profiles of urinary proteome between healthy control subject, patients with and without the LRRK2 gene mutation, for PD biomarker discovery.

#### 2.1.1. In Vivo, In Situ and In Vitro Salivary Pellicle Formation

Over the years, both in vitro and in vivo studies have been conducted to investigate salivary pellicle proteins. In situ salivary pellicle collection has recently been on the rise [13]. The majority of in vitro studies are carried out using hydroxyapatite or enamel slabs to form salivary pellicles with a pre-collected saliva sample, whereas in vivo salivary pellicle formation is usually carried out on freshly cleaned accessible tooth surfaces that pose a low risk of saliva contamination [4]. Notably, the major limitation of in vivo sample collection is that limited quantities of formed salivary pellicle can be harvested from tooth surfaces which prevented the extraction of this protein film with traditional biochemical methods, such as one- or two-dimensional electrophoresis [14]. Care should be taken for intraoral study site selection of in vivo salivary pellicles as salivary pellicle protein compositions appear to be region and site-specific [8]. A recent report by Trautmann et al. [13] suggested that perhaps in situ salivary pellicle collection may enable the identification of a high number of proteins, hence it is one good choice for extensive proteomic analysis. Despite a previous report claiming notable variances in amino acid compositions between in vitro on hydroxyapatite and in vivo formed salivary pellicle [7], the variables of in vitro studies are easier to control and the results generated are considered stable and reproducible.

#### 2.1.2. Sample Collection and Protein Extraction

Salivary pellicle proteins exist in small amounts (µg levels) [15], therefore removing the attached proteins from specimen surfaces without contamination is crucial for accurate analysis. Mostly, in situ salivary pellicle collection starts with professional dental prophylaxis, followed by refraining from eating and drinking prior to an experiment, then the selected teeth are isolated, water-rinsed, and air-dried [4]. Subsequently, the salivary pellicle is formed and harvested from the coronal two-thirds of buccal and labial/palatal surfaces of incisors and premolars by mechanical curetting [16], or mechanical scaling combined with chemical solubilisation [17]. Hannig et al. [18] compared 21 chemical and/or mechanical in situ salivary pellicle removal methods and concluded that most methods resulted in incomplete removal of the salivary pellicle layer, and the combination of ‘sponge-rubbing’ and ethylenediaminetetraacetic acid (EDTA) treatment was recommended as a stepwise method for salivary pellicle removal. Recently, the same group attached bovine enamel to human subjects to form a salivary pellicle, detached and removed residual saliva and non-adsorbed epithelial cells or microorganisms and air-dried. The subsequent extraction of the adsorbed pellicle components was achieved with an EDTA-free protease inhibitor mix [13].

Intraoral appliances with mounted specimens for in vivo salivary pellicle studies were introduced by Turner [19]. Sectioned extracted teeth were placed in a removable partial denture, from which the in vivo formed salivary pellicle was studied. Later, methods were developed to use pre-collected saliva to form in vitro salivary pellicles, known as ‘experimental pellicles’, on extracted teeth or material surfaces [20]. The subsequently formed salivary pellicle proteins could be collected using a dissociating agent through vortexing, sonication, centrifugation [15] or electrode filter papers [21].

### 2.2. Complex Protein/Peptide Separation

To date, single-step analytical techniques or instruments, capable of identifying and quantifying complex protein samples such as salivary pellicle are not available [22]. Multiple technologies are thus combined to separate, identify and quantify the protein mixtures. Current prevailing methods to identify proteins are “bottom-up” proteomics, which achieves protein identification mostly by mass spectrometry after enzymatic and/or chemical cleavage of protein-originating peptides, and chromatographic/electrophoretic strategies to decrease peptide mixture complexity [22]. Out of the bewildering variety of available techniques and instruments for protein separation, two main routes, gel-based approaches and gel-free approaches are outlined.

#### 2.2.1. One and Two-Dimensional Electrophoresis

One-dimensional electrophoresis, also known as sodium dodecyl-sulphate polyacrylamide gel electrophoresis (SDS-PAGE) separates denatured protein samples by molecular mass, which enables subsequent in-depth protein analysis by mass spectrometry, and simultaneously filtering out the low molecular weight impurities [23]. Two-dimensional gel electrophoresis (2-DE) separates complex proteins in two dimensions, isoelectric point and molecular mass, which is capable of separating several thousand proteins in one single gel piece [24]. Protein identification is achieved through analysing isolated gel spots with mass spectrometers, such as matrix-assisted laser desorption ionization-time of flight mass spectrometry (MALDI-TOF-MS) or liquid chromatography–tandem mass spectrometry (LC-MS/MS) [25].

Presently, the applications of gel-based protein separation are greatly reduced in use with the development of mass spectrometry (MS) [24]. Even though gel-electrophoresis pre-separation of target proteins in line with the analysis of in situ digested proteins by liquid chromatography–mass spectrometry (LC-MS) are known as gel-enhanced liquid chromatography–mass spectrometry, this is still considered one of the standard procedures for in-depth protein characterization of complex protein mixtures [23].

#### 2.2.2. Gel-Free Approaches

Liquid chromatography-based separation techniques involve physical separation of peptide fragments obtained by protease digestion of protein extracts and often coupled with mass spectrometry analysis [26]. In liquid chromatography, the digested peptide mixture is dissolved in a liquid and forced by high pressure through a column packed with a stationary phase composed of irregularly or spherically shaped particles for certain peptide separations [27]. For instance, reverse-phase liquid chromatography (RP-LC) consists of silica beads, 3–5 µm in diameter, with alkyl chains of eight (C8) or eighteen (C18) carbon-organic-modified particles attached as the stationary phase. It is the most frequently used liquid chromatography to couple with MS [28]. To achieve high separation resolution and efficiency, RP-LC is usually combined with other types of chromatographic separation approaches to gain multidimensional protein resolving power. Among all different combinations, the most common ones are two-dimensional chromatography and three-dimensional chromatography [29]. Two-dimensional chromatography was developed by Link et al. [30], who subjected the digested protein mixtures separately into two independent chromatography phases, strong cation exchange and RP-LC, in a biphasic microcapillary column. Whereas, in three-dimensional chromatography, an extra avidin column is added between the cation-exchange chromatography and RP-LC to gain better coverage [31]. Compared to gel electrophoresis, the liquid phase separation method greatly reduces sample handling time allowing rapid result generation.

Advanced protein identification techniques with multidimensional protein separation approaches, known as multidimensional protein identification technology (MudPIT), are automated high-throughput protein identification methods and have been widely used in large-scale protein extraction from various organisms [32]. Fang et al. [33] identified 1479 proteins in human whole saliva using MudPIT coupled with electrospray ionization-tandem mass spectrometry (ESI-MS/MS). Moreover, MudPIT could detect proteins at very low levels [22], and thus has opened many possibilities for further investigation of salivary pellicle proteins.

### 2.3. Salivary Pellicle Protein Identification

Subsequent to salivary pellicle sample collection and separation, researchers must select the appropriate procedure for peptide fragmentation and identification. The commonly adopted instrumentation systems for salivary pellicle protein identification are MALDI-TOF-MS [34] and ESI-MS/MS [35]. Generally, MALDI-TOF-MS is usually coupled with 2D gel electrophoresis or pre-separated protein samples, whereas, ESI–MS/MS, a more sensitive and direct method, could couple with both gel electrophoresis and liquid chromatography, and enable identification of unambiguous proteins and large scale protein complexes [36]. With the advance and refinement in mass spectrometry, more hybrid systems have been introduced to enable more reliable protein identification, high-throughput, tolerant to salt, and lower cost [32]. For instance, the development of nano-ESI-MS/MS has greatly reduced sample consumption and enhanced mass spectrum resolution and mass detection accuracy [37]. Siqueira, et al. [38] combined SDS-PAGE with nano-ESI-MS/MS and identified 130 enamel surface salivary pellicle proteins. Recently published work also adopted NuPAGE with Nano-LC-HR-MS/MS and identified 1032 salivary proteins, among which 498 were found on teeth 3 min after cleaning [18].

The mass spectrum rummaging for protein identification could be achieved using different strategies, such as MS/MS database searching [39], amino acid sequence tag searching [40], and accurate mass and time tag searching [41], with the assistance of various database search engines such as Uniprot, Sequest, Mascot and MaxQuant. The common underlying statistics for protein identification are through possibility estimation of a detected mass spectra randomly matched to similar molecular weight sequences in a database, followed by calculating how close is a match, and allocated the statistical confidence to the matched proteins [42].

### 2.4. Salivary Pellicle Protein Quantification

Tooth surface salivary pellicle protein quantification between two or more physiological states would be of great importance but remains a challenge. The electrophoresis gel staining approach helps differentiate the quantities of multiple proteins present in a single 2-DE gel, however, the low level of salivary pellicle proteins largely limits their visualization on 2-DE maps [43]. Whereas, LC-MS based quantification approaches with higher sensitivity and efficiency, and better coverage than gel-based quantification have been the methods of choice in the past [22].

Most LC-MS based quantifications are applied in relative quantification of protein or peptide abundance between samples of interest, and the basic strategies fall into two broad categories: those that create stable isotope labelling tags on protein/peptides that could later be analysed by mass spectrometry, and the label-free methods that are based on the comparison of related peak intensities or spectral counts between MS runs (Figure 1) [44].

#### 2.4.1. Quantitative LC-MS with Labelling Strategies

The salivary pellicle is only amenable to post-biosynthetic labelling for quantitative LC-MS, which leaves two broader categories of in vitro labelling methods with chemical or enzymatic derivations [45]. Despite the differences in tags and platforms, both labelling methods follow the same general principle whereby the quantitative information extraction relies on the ratio of signal intensity between related stable isotope tagged peptide pairs or groups based on their chemically identical analyte profiles but different mass [46]. The two commonly adopted labelling methods are outlined.

The first robust and universal chemical labelling method developed for LC-MS quantification is an isotope-coded affinity tag (ICAT), which selectively alkylates protein lysis with chemical reagents that contain a linker region with a biotin tag of either none (D_0_) or eight deuterium atoms (D_8_) [47]. The successfully labelled peptides would contain enriched monomeric avidin-agarose, and the quantification is achieved by calculating the ratio of the peak areas labelled by D_0_ and D_8_ ICAT reagent, and the relative abundance of peptides and the corresponding proteins in both samples can be determined. Siqueira, et al. [48] detected the relative abundance change in several major proteins during salivary pellicle formation on enamel with the ICAT approach. However, the major drawbacks of ICAT include quantification errors in RP-LC mass spectrometry, additional clean-up steps to remove cleaved biotin prior to LC-MS, and selective detection of proteins with a high cysteine content [49].

After isobaric tags for relative and absolute quantitation (iTRAQ) labelling method were introduced, it gained popularity as an alternative to ICAT with simplified analysis and increased analytical precision and accuracy [49]. The most common label of this technology is 4-plex amine-reactive isobaric tags, which derive peptides at the primary amines of the N-terminus and C-terminal lysine side chains, and thereby labelling all peptides in a digest mixture [50]. The iTRAQ tags consist of three functional groups, the reporter group, balance group and peptide reactive group. The reporter group tends to fall off upon fragmentation and generates ions of different molecular mass, and the quantification information could be gained through analysis of reporter ions in MS upon integration of their relative abundance (Figure 2) [29], then statistically assemble them into protein ratios, which can then be computed and evaluated using related software platforms, such as Mascot and MaxQuant [51].

#### 2.4.2. Label-Free Quantification

As an alternative to labelling quantification, label-free quantification has wider application with lower expense, is less time consuming and not constrained to a limited number of group comparisons, which makes it a preferable method, especially in large sample size clinical proteomic evaluations [45]. The label-free approach could be further divided into two categories according to the evaluation method: those based on peak intensity or spectral counting (Figure 1B).

The peak intensity method achieves relative quantification through calculating peak intensity/area under the curve using the chromatographic signal intensity of peptide fragment peaks in MS. Typically, the mass to charge ratio (*m*/*z*) of peptide fragments are measured at a specific retention time in a sensitivity concentration range [24]. The chromatographic elution profiles of one or more peptide masses could be constructed from measurements, and the results produced are further integrated and serve as quantitative measurements of the original sample peptide concentrations. Theoretically, all ion signals within the sensitivity range of the MS analyser could be detected and integrated as part of the quantification process, however, unavoidable biological and systemic variance in retention time across sample groups is the major challenge for ion intensity evaluation between MS runs, especially in large peptide mixtures with wide-range ion signals [46]. Moreover, as full scan MS-level detection is required to distinguish peptide ion signals from various background chemical noises, MS platforms, i.e., triple-quadruple MS platforms, with dynamic range peptide detection, high-resolution power and mass accuracy are mandatory [46].

Spectral counting is an empirical observation that selects and measures spectra of peptides in a sample that matches specific proteins, and the subsequent relative quantification is achieved by comparing the frequency of the spectra between experiments [45]. Spectra counting is relatively simpler in statistical analysis and more reproducible with a larger detection range than the peak intensity approach [52]. It has certain advantages over isotope labelling, as the acquired mass spectra data in the chromatogram could be used for both protein identification and quantification [51]. However, the major issue that makes spectra counting a controversial approach is that it is based on the assumption that all proteins have the same spectral count response, and in fact, the inherent physicochemical properties of each peptide render a different chromatogram behaviour, and thus the method results in bias quantification outcomes [45]. Delius et al. [53] adopted a label-free quantitative nano-LC–MS/MS approach and identified 72 major proteins in the initial pellicles formed intraorally on dental ceramic specimens already after 3 min with high inter-individual and inter-day consistency.

### 2.5. Salivary Pellicle Protein Functional Characterization

The next step of the proteomic workflow lies in the functional interpretation of identified proteins. With the advance of bioinformatics and system biology, biologists and clinical researchers are confronting a plethora of methodologies in functionally characterizing proteins of interest. Generally, the choice of technique should be based on the aim of the study and the nature of the protein samples.

#### 2.5.1. Integration to Functional Proteomics/Proteogenomics

Gene encoding proteins, and thus the inherent genome sequences and expression profiles could enable the elucidation of functional linkages between proteins [54]. In the post-genomic era, advanced computational proteomics approaches could assist to extract this knowledge.

One of the most common solutions to aid protein data interpretation is Gene Ontology (GO) curation, which is developed for standardized large-scale protein annotations through assigning a structured GO vocabulary set to well-characterized proteins [55]. Three general categories have been assigned to the hierarchical GO structure to construct a descriptive framework protein annotation, namely the biological process that a protein contributes towards, the molecular function that a protein possesses and the cellular component where a protein is active [56]. For instance, GO annotations for salivary protein transforming growth factor beta-1 includes the biological process of ‘protein phosphorylation’ (GO: 0006468), the molecular function of ‘transforming growth factor beta receptor binding’ (GO: 0005160), and cellular component of ‘extracellular space’ (GO: 0005615). Depending on the availability of the annotation database, one protein could be assigned to multiple GO terms under these three general domains, and as each GO term may have multiple connections with broader ‘parent’ terms and more specific ‘child’ terms, GO annotation could facilitate a broad understanding of functional attributes and extended investigations of a data set [57].

#### 2.5.2. Protein Function Prediction Based on Network Analysis

The advance of high throughput proteomics has resulted in an enormous amount of data, hence, the traditional methods of investigating individual proteins are no longer adequate for incorporating intricate molecular details of dynamic pathophysiological status. The computational approaches are thus developed to bridge the gap between data acquisition and interpretation.

As most proteins exert their functions through interactions with other proteins [32], one way to characterize the functional aspects of proteins is to seek their interaction partners. The generation of protein-protein interaction data could be achieved from both experimental technologies and computational detection. The most commonly used experimental approach is the two-hybrid system [58], which is based on the premise that a pair of upstream hybrid interaction proteins with transcription factors incorporated could activate downstream transcription domains, and further regulate an adjacent reporter gene. Ito, et al. [59] identified 4549 protein-protein interactions among 3278 proteins using a two-hybrid approach. Alternatively, bioinformatics-aided detection of relevant protein interactions could be achieved through analysing the established genomes, that if two proteins share an identical or similar pattern of presence or absence in all surveyed genomes (orthologs), these two proteins could be inferred as having a functional link, or existing in a common pathway or complex [54]. Other computational approaches include analysing the fusion fashion of protein domains and the gene neighbour method, both of which are based on the premise that genetic similarities between gene products suggest functional similarity [60].

As more protein interactions are identified, two-dimensional protein networks can be constructed with the assistance of software platforms, such as GeneMANIA and STRING, to visualize the dynamic interactions between proteins of interest (Figure 3). As mentioned above, protein function prediction could be achieved through their interaction partners in a network, once a function has been assigned to one protein (i.e., through GO annotation), it could be inferred that the proteins linked to this protein in the network may possess a similar function or be present in the same pathway or complex [61]. Hence, the information gained through analysing protein interaction networks is of great value for simulating the formulation of further hypotheses or adding additional information to the existing knowledge infrastructure [54]. The incorporation of functional network in the characterization and regulation of disease mechanisms has been attempted. von Scheidt et al. [62] adopted tissue-specific functional networks to identify and characterize central regulators and networks leading to atherosclerosis.

#### 2.5.3. Differential Expressed Proteins

For most of the investigations, the information of differential protein expression in various biological conditions is of great value to gain insights into underlying disease pathogenesis. However, the unprecedented high-resolution data generated by recent advanced high throughput mass spectrometry put great challenges in acquiring correspondingly high-resolution data annotation. Currently, applications of computational strategies in proteomics could address this question in different annotation levels, ranging from a global view of all the available interactions to highly formalized pathways. A variety of online resources, such as DAVID (https://david.ncifcrf.gov/, accessed on 28 February 2021), KEGG (http://www.genome.jp/kegg/pathway.html, accessed on 28 February 2021), and Reactome (http://www.reactome.org, accessed on 28 February 2021), are available for users to search their protein data for any statistical enrichment in a given biological phenomenon to gain insights into the biological condition being studied. For instance, Xie et al. [63] tested the diagnostic potential of cells present in saliva, with numerous proteins identified in oral squamous cell carcinoma signalling and tumorigenesis pathways. However, the independence of different databases result in cumbersome results, the inability to model dynamic states of a system, and the inability to consider interactions between pathways still address the needs for high coverage, resolution, and accuracy of knowledge-based annotations [64].

## 3. Challenges and Future Perspectives

Mammalian saliva carries a precise developmental and biological program that defines the function of multiple organs. Saliva is at the interface of dynamic interactions between microbial ecosystems and oral immune cells [1]. Dysregulation of saliva physiology has an impact on oral and systemic health, which makes it a potential tool for the diagnosis of oral and systemic diseases [65]. On the other hand, an altered systemic state leading to changes in salivary gland physiology and/or functions, or directly, the formed salivary pellicle, could alter the proteome of the latter hence increase the susceptibility of the affected hosts to oral and dental diseases [66,67].

For instance, in patients who suffered from gastroesophageal reflux and erosive tooth wear, in-depth proteomic profiling of the affected salivary pellicles revealed distinct protein profile, especially membrane-binding proteins like lysozyme C, antileukoproteinase, cathepsin G, neutrophil defensins and basic salivary proline-rich proteins while subunits of haemoglobin, albumin and isoforms of cystatin appeared enriched [66]. In head and neck cancer patients treated by irradiation therapy, proteomic investigation showed that statherin, one of the putative key proteins in salivary pellicle formation, was markedly increased in salivary pellicle during radiation therapy while innate defence protein like lactotransferrin, proline-rich proteins, cystatins, neutrophil defensins 1 and 3 and histatin-1 were decreased. After radiotherapy, lactotransferrin appeared to increase, while statherin was decreased. Interesting, the tumour suppressor protein: modulator of apoptosis 1 protein was detectable exclusively in salivary pellicle after radiotherapy [67]. The authors concluded that in head and neck cancer survivors after radiotherapy, their salivary pellicle proteome was remarkably altered. Both association studies provided insights into possible disease pathogenesis that clinicians/scientists could consider explore further the mechanism or aetiology for the respective ailment.

Despite implications of findings from the above two studies by the Brazilian group of Buzalaf are yet to be determined, they showcased the utilities of contemporary salivary pellicle proteomic approaches alongside deciphering the biological relevance of salivary pellicle in health and disease. Interesting, a report by Isola et al. [68] reported in submandibular gland biopsies of type 2 diabetic patients, statherin expression in secretary granules of serous cells were significantly reduced. The authors hypothesized that the corresponding salivary pellicle of type 2 diabetic patients might potentially be affected hence increasing the latter’s susceptibility to oral infection. Follow up proteomic studies of the salivary pellicle of type 2 diabetic patients are needed to verify such postulation.

As evidenced from research reports investigating effects of irradiation therapy or gastroesophageal reflux etc. on salivary pellicle, the minimally invasive collection, easy access, and the negligible risk to the donor of saliva and/or saliva pellicle proteomic investigation possess major advantages over other biofluids, such as blood, [69], which may make it more readily applicable to special needs populations.

The complexity of the oral environment, the dynamic protein adsorption process and relatively low protein quantities available for extraction and characterization greatly limit investigations of tooth surface salivary pellicle. Mass spectrometry-based proteomics has become the method of choice for protein complex analyses, yet the technical and inter-individual variations, the comparison of the datasets of the same salivary pellicle proteome between laboratories could result in less agreement [70]. Moreover, the currently available information from high throughput proteomic experiments enable the construction of large scale protein networks for automatic protein function prediction and annotation. However, the lack of standard protocols for data processing, database searching and data sharing among different databases, platforms and software [49] create more gaps and challenges in salivary pellicle protein studies. Therefore, the standardization of data formats and construction of related protocols are essential for further salivary pellicle protein investigations.

## 4. Summary

The dynamics of the oral environment and limited amounts of protein that can be collected in vivo pose great challenges for salivary pellicle proteomic endeavours. To date, reproducible and consistent salivary pellicle protein identification and comprehensive characterization are rather scarce and remain protracted. Tremendous progress has been made in the field of proteomics and bioinformatics in the past decade and has greatly enlarged the original scope of protein analysis, which is not only limited to simple cataloguing of protein species but also can characterize proteins in a functional context with a deeper insight into processes in combination of proteomic data with microbiomics, and metabolomics data. Now with the assistance of promising methods and technologies, a revolutionary advance is likely in proteomic investigations of salivary pellicle and other low-volume highly-complex protein samples such as gingival crevicular or peri-implant sulcular fluids.

## Figures and Tables

**Figure 1 jcm-10-02801-f001:**
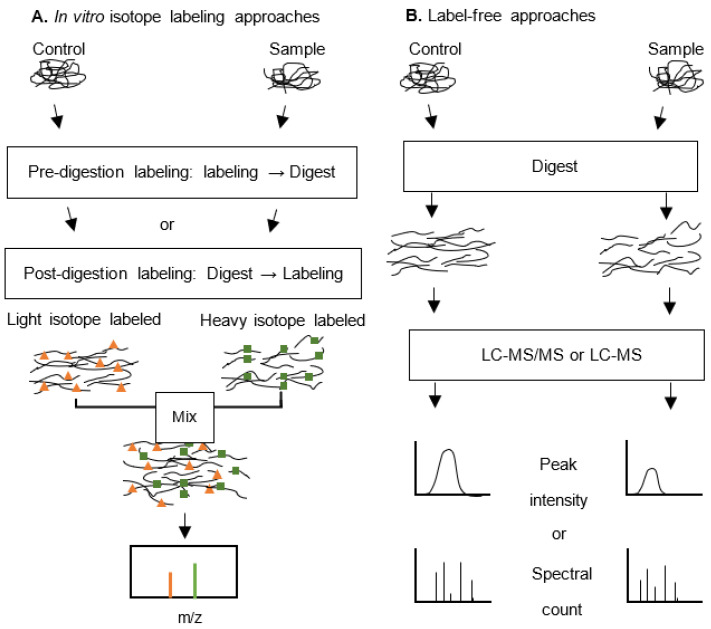
Overview of proteomic quantification approaches. (**A**) Stable isotope labelling method. After labelling with a stable isotope of different molecular mass, the samples are combined and analysed by mass spectrometry. The relative quantification is achieved through calculating the intensity ratio of labelled peptide pairs/groups [mass to charge ratio (*m*/*z*)]. (**B**) Label-free quantification approach. Samples are digested and subject to liquid chromatography–tandem mass spectrometry (LC-MS/MS) analysis individually. The subsequent quantification is based on the comparison of peak intensity or spectral count of the peptide or the protein. The stable isotope labelling allows for more accurate quantification of protein expression, whereas label-free methods could be employed to avoid certain drawbacks of the former, such as inefficient labelling, difficulty in the analysis of low abundance peptides, and sample number limitation (cf. [24]).

**Figure 2 jcm-10-02801-f002:**
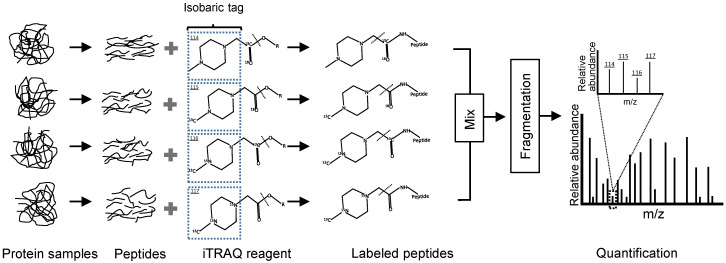
Common 4-plex isobaric tags for relative and absolute quantitation (iTRAQ) workflow. The reporter group (isobaric tag) tends to fall off upon fragmentation and generates ions of different molecular mass (i.e., *m*/*z* from 114 to 117), and common to other isobaric and isotopic labelling, is the downstream processing where the labelled peptide ratios are computed, and then statistically assessed to evaluate the assembled protein ratios (cf. [51]).

**Figure 3 jcm-10-02801-f003:**
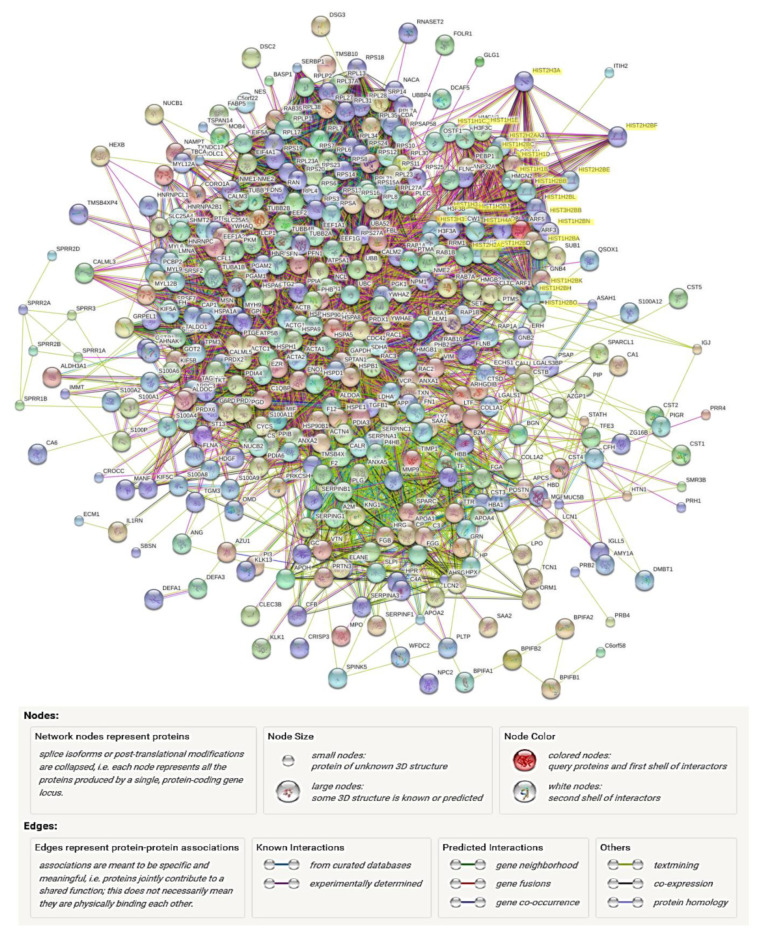
Visualization of protein interaction network of experimental in vitro tooth surface salivary pellicle proteins (unpublished data). The complex experimental protein data could be submitted to various software platforms to visualize the dynamic interactions between the identified proteins. Through network demonstration, some proteins are found possessing dynamic connections with many other proteins, such as histones (HIST, highlighted yellow). However, the complication of global networks limits the direct information visualizations, thus further simplifications may be needed to gain better resolutions. Image generated by STRING platform (http://stringdb.org, accessed on 28 February 2021) through searching genomic associations between genes.

## Data Availability

The authors elect to not share the data.

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
