# Peer review of "Analysing Complex Oral Protein Samples: Complete Workflow and Case Analysis of Salivary Pellicles"

_jcm, 2021, doi:10.3390/jcm10132801_

Round 1

Reviewer 1 Report

The reviewer highly appreciates papers on the pellicle layer. The manuscript is well written, nice images illustrate the main aspects.

However, some aspects should be reworked precisely/ added:

Acquisition of samples: please compare precisely advantages and disadvantages of different sample acquisition: in vivo vs. in situ vs. in vitro

Conclusions and perspectives of future research: it should be pointed out that deeper insight into the processes in the oral cavity is only possible, if proteome data are evaluated together with micobiomics and metabolmics data.

Please discuss the high intraindividual and interindividual variablility of initial bioadhesion. This limits the applicability as a biomarker.

Please consider other recent papers in this field of research concerning methods and outcome:

 Deep Proteomic Insights into the Individual Short-Term Pellicle Formation on Enamel-An In Situ Pilot Study.

Trautmann S, Künzel N, Fecher-Trost C, Barghash A, Schalkowsky P, Dudek J, Delius J, Helms V, Hannig M.Proteomics Clin Appl. 2020 May;14(3):e1900090. doi: 10.1002/prca.201900090. Epub 2020 Apr 28.PMID: 32237277

Label-free quantitative proteome analysis of the surface-bound salivary pellicle.

Delius J, Trautmann S, Médard G, Kuster B, Hannig M, Hofmann T.Colloids Surf B Biointerfaces. 2017 Apr 1;152:68-76. doi: 10.1016/j.colsurfb.2017.01.005. Epub 2017 Jan 6.PMID: 28086104

Maybe, some studies of the grandmaster in pellicle proteomics Walter Siqueira have been overlooked. Please check carefully.

Author Response

Comments to Authors

  • “The reviewer highly appreciates papers on the pellicle layer. The manuscript is well written, nice images illustrate the main aspects.”
  1.  

Authors’ reply:

We appreciate very much Reviewer 1’s support regarding this manuscript.

“However, some aspects should be reworked precisely/added:”

  1. “Acquisition of samples: please compare precisely advantages and disadvantages of different sample acquisition: in vivo vs. in situ vs. in vitro

Authors’ reply:

We thank Reviewer 1’s suggestion, please note that related details were now added under Section 2.1.1, in particular with reference to relevant publications by Drs. Siqueira and Hannig’s groups (lines 94-102).

  1. “Conclusions and perspectives of future research:
  2. it should be pointed out that deeper insight into the processes in the oral cavity is only possible, if proteome data are evaluated together with microbiomics and metabolomics data.”

Authors’ reply:

Totally agree. The Conclusion Section is revised to highlight the important concept related (lines 420-421)

  1. “Please discuss the high intraindividual and interindividual variability of initial bioadhesion. This limits the applicability as a biomarker.”

Authors’ reply:

Thank you very much for the suggestion. Section 2.1 (last 2 sentences) was revised to highlight such facts.

  1. “Please consider other recent papers in this field of research concerning methods and outcome:

Trautmann S, Fecher-Trost C, Barghash A, Schalkowsky P, Dudek J, Delius J, Helms V, Hannig M. Deep Proteomic Insights into the Individual Short-Term Pellicle Formation on Enamel-An In Situ Pilot Study. Proteomics Clin Appl. 2020 May; 14(3):e1900090. doi: 10.1002/prca.201900090. Epub 2020 Apr 28.PMID: 32237277

Delius J, Trautmann S, Médard G, Kuster B, Hannig M, Hofmann T. Label-free quantitative proteome analysis of the surface-bound salivary pellicle. Colloids Surf B Biointerfaces. 2017 Apr 1;152:68-76. doi: 10.1016/j.colsurfb.2017.01.005. Epub 2017 Jan 6.PMID: 28086104

Authors’ reply:

Our apologies for the omission. We now referred to more appropriate recent papers in the revised manuscript (References 11-14, 21, 53 & 62)

  1. “Maybe, some studies of the grandmaster in pellicle proteomics Walter Siqueira have been overlooked. Please check carefully.”

Authors’ reply:

Again, our sincere apologies for the omission. We added 2 more relevant references from Dr. Siqueira’s group [references 11 & 14].

Reviewer 2 Report

The present review is very interesting, contains an actual topic and is well-written. Few points should be considered, as mentioned bellow.

In the introduction, the authors mention other terms that refer to “salivary pellicle”, such as “dental pellicle” or “acquired enamel pellicle”. Throughout the text these different terms are used, but it would be better/clearer if the authors use only one term throughout the text, for example “salivary pellicle”, as used in the title.

Lines 56–61: “As the oral cavity is a unique portal where mineralized tissue is directly exposed to heavy colonization of oral microbiota, it is plausible that the acquired pellicle interface between tooth surfaces and the oral environment holds essential biological functions, such as protection from erosion due to dietary habits or gastroesophageal reflux disease, demineralization, lubrication, and intimate relationships with oral microbes [8]”.

This sentence should be revised, as it mixes oral microbiota, which is connected to dental caries, with dental erosion, which is not related to microbiota, making it confusing.

What about the UniProt database?

Most of the references are older then 10/20 years. It would make the manuscript even more interesting if examples of the methods used nowadays are mentioned. For example, lines 353–355, 369. Accordingly, it is not clear in the manuscript that “Tremendous progress has been made in the field of proteomics and bioinformatics in the past decade”.

Author Response

“The present review is very interesting, contains an actual topic and is well-written.”

Authors’ reply:

We appreciate very much Reviewer 2’s support regarding this manuscript.

“Few points should be considered, as mentioned below.”

1. “In the introduction, the authors mention other terms that refer to ‘salivary pellicle’, such as ‘dental pellicle’ or ‘acquired enamel pellicle’. Throughout the text these different terms are used, but it would be better/clearer if the authors use only one term throughout the text, for example ‘salivary pellicle’, as used in the title.”

Authors’ reply:

Very much appreciate the comment, indeed the various terms used could create confusion. We have revised the text and used ‘salivary pellicle’ as the only one term throughout the paper, if appropriate.

2.“Lines 56–61: ‘As the oral cavity is a unique portal where mineralized tissue is directly exposed to heavy colonization of oral microbiota, it is plausible that the acquired pellicle interface between tooth surfaces and the oral environment hold essential biological functions, such as protection from erosion due to dietary habits or gastroesophageal reflux disease, demineralization, lubrication, and intimate relationships with oral microbes [8]’.

This sentence should be revised, as it mixes oral microbiota, which is connected to dental caries, with dental erosion, which is not related to microbiota, making it confusing.”

Authors’ reply:

Our apologies for the confusion. The sentence is now revised focusing on caries protection only (lines 56-63)

3. “What about the UniProt database?”

Authors’ reply:

Our apologies for the omission, indeed UniProt database was also referred to and the omission was rectified (line 204).

4. “Most of the references are older than 10/20 years. It would make the manuscript even more interesting if examples of the methods used nowadays are mentioned. For example, lines 353–355, 369. Accordingly, it is not clear in the manuscript that ‘Tremendous progress has been made in the field of proteomics and bioinformatics in the past decade’.”

Authors’ reply:

Totally agree with Reviewer 2. We have added references describing new frontier work e.g. Max Planck Institute (lines 84-87; 365-368), Hannig’s group [13] (lines 119-122), and Buzalaf’s group [21] (line 129), etc.

Reviewer 3 Report

The present manuscript is a simple narrative review on the topic. It is general and covers the topics that it set out to discuss.

Only a minor point the authors could consider:

Line 100. A new paper from Prof Hannig and his group already use another method for in situ pellicles. Please have a look at: https://doi.org/10.1002/prca.201900090

And Line 108. There is also more recent paper showing the differences between methods for collecting and analysing in vitro pellilces. Please take a look at: http://dx.doi.org/10.1590/1678-7757-2020-0189

Author Response

1. “The present manuscript is a simple narrative review on the topic. It is general and covers the topics that it set out to discuss.”

Authors’ reply:

We appreciate Reviewer 3’s comment regarding this manuscript.

“Only a minor point the authors could consider:”

2. “Line 100. A new paper from Prof Hannig and his group already use another method for in situ pellicles. Please have a look at: https://doi.org/10.1002/prca.201900090”

Authors’ reply:

We thank Reviewer 3 regarding the heads-up on recent publication from the Hannig’s group. We updated the revised manuscript accordingly, citing the contemporary works. (line 119-122)

3. “And Line 108. There is also more recent paper showing the differences between methods for collecting and analysing in vitro pellilces. Please take a look at: http://dx.doi.org/10.1590/1678-7757-2020-0189”

Authors’ reply:

We appreciate Reviewer 3 for the critique regarding methodology concerning collection and analyzing in vitro pellicles. We have now revised Section 2.1.2 accordingly (line 129).

This manuscript is a resubmission of an earlier submission. The following is a list of the peer review reports and author responses from that submission.